

# Impacts of climate change on extreme floods in Finland –studies using bias corrected Regional Climate Model data

Noora Veijalainen[1], Juho Jakkila[1], Taru Olsson[2], Leif Backman[2], Bertel Vehviläinen[1], Jussi Kaurola[2]

[1]Freshwater Centre, Finnish Environment Institute, Mechelininkatu 34a, P.O. Box 140, FI-00251, Helsinki, Finland
[2]Finnish Meteorological Institute, Erik Palménin aukio 1, P.O. Box 503, FI-00101, Helsinki, Finland

*Correspondence to*: Noora Veijalainen (Noora.Veijalainen@ymparisto.fi)

**Abstract.** Bias correction of precipitation and temperature of five Regional Climate Models (RCMs) was carried out using Distribution Based Scaling (DBS) method with two versions for precipitation adjustment: single gamma and double gamma. This data were then used as input for a hydrological model to simulate changes in floods by the end of this century, and the results were compared to corresponding changes simulated using delta change approach. The results show that while the DBS adjustment significantly improves the RCM precipitations and temperatures compared to observations, especially the double gamma distribution does not always preserve trends of the uncorrected RCM data. The simulation of floods in the control period is improved by the DBS adjustment with no significant differences between single and double gamma. However, some scenarios are still unable to match the observed hydrology adequately due to remaining biases especially in near zero winter temperatures. These scenarios may produce an unrealistic climate change signal and should therefore be discarded from further use. A simple criterion for evaluating the adequate performance of the RCMs and hydrological models compared to observed floods is presented. The results of climate change simulations show that extreme summer precipitations increase more than average values in Finland. The changes in floods by 2070–2099 vary in different regions depending on season and the main flood producing mechanism (snowmelt or heavy rain). The changes in floods simulated with the DBS adjusted RCM data are mostly similar as with delta change approach, but the DBS method produces larger range of changes.

## 1       Introduction

Extreme precipitation events and floods can potentially cause significant damage to properties and industries and disruption to critical functions of society such as transport. The extremes of Finnish weather are moderate compared to other parts of the world and Finland is considered one of the least vulnerable areas for flooding in Europe (Schmidt-Thomé et al., 2006). However extreme floods still have potential to cause large damages, a flood with return period of 250 years could cause damages of 710 million euros (Ollila et al., 2000). Climate change is projected to alter extreme precipitation amounts and flood magnitudes, and evaluating these potential changes is important to enable adaptation to climate change impacts.



Adaptation through land use planning, flood mapping, flood risk management and changes in regulation rules and practices can help diminish potential increases in flood damages.

Climate scenarios for Finland project an increase in temperature of 2.4–6.0 °C by 2070–2099 compared to 1971-2000 (Jylhä et al., 2009a; Ruosteenoja, 2013) and an increase in precipitation of 8–21 %. Heavy precipitation events have been estimated
to increase, in most cases more than the average values (Lehtonen, 2011; Jylhä et al., 2009b). The projected changes in climate have both increasing and decreasing influences on floods in Finland. Decreasing snow accumulation during warmer winters can cause decreases in spring floods, while increase in precipitation and milder winters can cause increases especially in autumn and winter floods (Veijalainen et al., 2010).

Conventional methods used to estimate climate change impacts on hydrology include hydrological modelling using the delta
change method, where observed climate is changed according to climate scenarios and a hydrological model is used to simulate the discharges and water levels in this perturbed climate (e.g. Arnell, 1999; Hay et al., 2000). This method has several limitations especially with regards to estimating climate change impacts on extremes such as floods. In the most used version of the delta change method the monthly mean changes in temperature and precipitation are used to modify the observed temperatures and precipitations (e.g. Arnell, 1999; Hay et al., 2000). Changes in extremes are not considered
separately, even though changes in extreme precipitation may be crucial for floods especially in small watersheds. Alternative methods, such as use of Regional Climate Model (RCM) daily data as input for the hydrological model, have recently become more common (e.g. Teutschbein and Seibert, 2012; Vormoor et al., 2015). These methods use more of the climate model data and can represent changes in extremes and in spatial scale better than the simpler methods.

However, the RCM data still contain significant biases when compared to observations (Christensen et al., 2008;
Teutschbein and Seibert, 2012). The direct use of this data as an input of the hydrological models leads to considerable misrepresentation of current hydrology and potentially misleading results of climate change impacts on hydrology (Wood et al., 2004). The bias correction of the daily RCM precipitations and temperatures is therefore necessary to produce seasonal and regional hydrological variability, and extremes, that are in line with the observations of the control period. Several methods to correct the RCM biases have been developed and compared (Berg et al., 2012; Chen et al., 2011; Teutschbein
and Seibert, 2012). While simple bias correction methods correct the mean values sufficiently, they are often unable to correct the higher moments of the distribution (Berg et al., 2012; Chen et al., 2011; Teutschbein and Seibert, 2012). The higher moments are especially important when extremes, such as heavy precipitation and floods, are studied.

This paper is a continuation to the paper by Olsson et al. (2015), where the bias correction of RCM temperatures and precipitations was carried out with the Distribution Based Scaling (DBS) method and the results were analysed with focus on
changes in annual and seasonal mean values. This paper uses the same methods and models but in contrast will focus on extremes. The first aim of this article is to investigate the changes in extreme precipitation projected by RCMs in 2070-2099 and how bias correction based on observations affects these results. The second aim is to use the bias corrected RCM data as input to a hydrological model to estimate the impacts of climate change on floods. For this purpose hydrological model simulations are carried out in four case study locations and also on large selection (67) of catchments in different parts of





Finland (referred to in the text as national data set) to gain a broader view of the climate change influence on floods by 2070–2099. Comparison with results of the delta change method is carried out to examine the differences and advantages and disadvantages of both methods.

## 2    Materials and methods

In Finland floods are mainly caused by snowmelt, heavy precipitation, sea level rise or a combination of these. Also ice jam and frazil ice floods are fairly common. Spring snowmelt is an important cause for floods and the dominant cause in northern part of the country. The small and medium size coastal rivers in southern and western Finland have more mixed flood regimes, with rainfall also being an important cause for floods. In central and eastern Finland the hydrology is characterized by numerous interconnected lakes, which cause time lags and detention of water. Central lakes and their outflow rivers therefore have long lasting floods, which can take several months to accumulate.

### 2.1    Study catchments

In this study 67 catchments in different parts of Finland are simulated to enable overview of climate change impacts on flood risks in Finland (Fig. 1). The 67 catchments, hereafter referred to as national data set, were chosen to represent different types of catchments in Finland, prioritising discharge stations with long observation series. The same national data set of 67 catchments were used by Veijalainen et al. (2010) to estimate changes in floods with the delta change method. In addition four catchments are used as case study and example sites and examined in more detail.

The four catchments used as case study catchments are Lake Saimaa, Lake Nilakka, Loimijoki and Ounasjoki (Fig 1). These are the same areas as in previous article by Olsson et al. (2015), except Lake Saimaa is included instead of Lake Lentua, because it is a more important flood risk area. Loimijoki (Maurialankoski observation station, catchment area 2 650 km$^2$, lake percentage 3.1) is a medium sized coastal river with high proportion of cultivated areas on clay soils in the catchment area. Nilakka (catchment area 2 160 km$^2$, lake percentage 18) is a large lake in the lake area in central Finland, which is characterized by numerous lakes. Lake Saimaa is the largest lake in Finland, and the largest catchment in Finland located in the lake area of eastern Finland (catchments area 61 000 km$^2$, lake percentage 20). The outflow river of Lake Saimaa, Vuoksi River, is a transboundary river flowing to Russia, which influences flood management in the area. Ounasjoki (Marraskoski observation station, catchment area 12 300 km$^2$, lake percentage 2.6) is a medium size river in northern Finland (Fig. 1). The river has been kept in its natural state and its snowmelt floods occur in spring and early summer. (Korhonen and Kuusisto, 2010)

### 2.2    Data and bias correction

The observational and RCM data used here was the same as in Olsson et al. (2015) and included observations of temperature, precipitation, wind speed, humidity, water levels and discharge. Discharge observations from observation



stations of the national data set were also used for comparisons. The five climate scenarios used were from four different RCMs and four different GCMs: Table 1). These were obtained from the ENSEMBLES project data archive (van der Linden and Mitchell, 2009).

The bias correction method used to correct temperature and precipitation was the distribution based scaling (DBS) method (e.g. Yang et al., 2010; Teutschbein and Seibert, 2012). Several other names such as quantile mapping, quantile-quantile mapping, distribution mapping or statistical downscaling are used for this or similar approaches, where a transfer function is used to correct the distribution the RCM simulated variables (Deque et al., 2007; Piani et al., 2010; Teutschbein and Seibert, 2012). In the DBS method temperature is described by a Gaussian (normal) distribution with daily mean ($\mu$) and standard deviation ($\sigma$) and precipitation with gamma distribution with monthly shape ($\alpha$) and scale ($\beta$) parameters.

The DBS approach (Yang et al., 2010; Olsson et al., 2015) for temperature (T) included the following steps: To calculate seasonal mean and standard deviation these values were averaged using a 15-day moving window and were further smoothed with Fourier series with five harmonics on a daily basis over the control period (1961–2000). These smoothed daily mean and standard deviation for each grid point were then used to calculate the daily CDFs for observations, and for the control period. The DBS parameters for the control period were used also to adjust the scenario runs. The separate temperature distributions for dry and wet days (Olsson et al. 2015) were not used in this study. This decision was made since in Olsson et al. (2015) the separate corrections performed worse than the using one temperature correction due to the small fraction of dry days during some seasons. Results of Olsson et al. (2015) show that both temperature corrections produce very similar discharge simulations.

For precipitation (P) single and double gamma distribution were used. Two methods were used to estimate if separate correction of extreme precipitation will improve the simulations of extreme floods. For both distributions, excessive drizzle days in the RCM data were first removed by defining a cut-off value that reduced the percentage of wet days in the RCMs to match with the observations on a monthly basis. The remaining daily precipitation was adjusted to match the observed frequency distribution by using gamma distribution (single gamma). To better capture the extreme precipitation events also a double gamma distribution was used by separating the observed and RCM precipitation distribution into two partitions by the 95th percentile, resulting in two sets of parameters for the gamma distributions, one for the below 95th percentile precipitation and another set for above it. These monthly parameters for each grid point were then used to calculate the daily CDFs for observations and RCMs during the control period. Monthly DBS parameters for the control period and the monthly precipitation threshold were used also for the scenario runs.

The details of the method and the formulas used are described in Olsson et al. (2015). Olsson et al. 2015 provided the comparison of the bias corrected temperatures and precipitations with observed values, for average values, while this study will focus on extremes.



## 2.3    Hydrological modelling

The hydrological modelling in this study was carried out using the Watershed Simulation and Forecasting System (WSFS) operated and developed in Finnish Environment Institute (Vehviläinen et al., 2005). It is a HBV-type (Bergström, 1976) conceptual hydrological model covering entire Finland and transboundary watersheds. The WSFS is used as the national hydrological forecasting and flood warning system in Finland (Finnish Environment Institute 2016). It has also been used for research purposes (e.g. Veijalainen et al., 2012; Jakkila et al., 2014; Huttunen et al. 2015). More details about the model are found in Olsson et al. (2015) and Vehviläinen et al. (2005).

The WSFS hydrological model consists of small sub-catchments, numbering over 6 000 in Finland (Vehviläinen et al., 2005). The sub-models in WSFS include a precipitation model calculating areal value and form for precipitation, a snow accumulation and melt model based on the temperature-index (degree-day) approach, a rainfall-runoff model with three soil/groundwater storages, and models for lake and river routing.

The WSFS was calibrated against water level, discharge and snow water equivalent observations from 1981–2012. The Nash-Sutcliffe efficiency criterion $R^2$ (Nash and Sutcliffe, 1979) for the discharges in control period 1971–2000 in the four case study catchments was 0.84 for Loimijoki, 0.60 for Lake Saimaa (outflow observations smoothed to disregard the weekly regulation, however regulation still impacts observed outflows in some situations), 0.81 for Lake Nilakka, 0.87 for Ounasjoki. For Saimaa, the $R^2$ for inflow, which is relatively unaffected by regulation is 0.90. For the national data set the average $R^2$ value for daily discharges was 0.82 and the range was from 0.56 to 0.94.

Discharges in control and future periods were also simulated with the delta change method (Arnell 1999; Hay et al. 2000) to provide comparison for the BC method. In the delta change method the average monthly changes in air temperature (in ºC) and precipitation (in %) projected by the RCM scenarios for future time periods were added to (for temperature) or multiplied by (for precipitation) the observed values of temperature and precipitation, respectively, from the control period. A temperature-dependent component was included using seasonal linear transfer functions so that the temperature change was dependent on the original control period temperature (Andréasson et al. 2004). According to RCM results for Finland, cold days, especially in winter, will warm more than the average monthly temperature (Räisänen et al. 2004; IPCC 2007). The modification of the standard delta change method is used to account for this change in distribution of temperature.

## 2.4    Frequency analysis

Frequency analysis for precipitation, discharge and water levels was carried out to estimate the magnitude of extreme precipitations and rare floods with return period of 100 years. For precipitation a GEV (Generalized Extreme Value) distribution is used and it is fitted to annual or seasonal daily maximum values of the catchments average precipitations. For discharges and water levels the Gumbel distribution was used since it most commonly used and officially recommended for flood frequency analysis in Finland (Ministry of Agriculture and Forestry, 1997). The GEV distribution was also tested for the national data set discharges simulated with observed temperatures and precipitations for the control period. The



assumption that the GEV coefficient of skew can be zero, and therefore Gumbel distribution can be used instead of GEV distribution, was tested with the likelihood ratio test (Coles, 2001). The test showed that for only 6.0 % of the catchments in the national data set this assumption was not accepted with 5 % confidence level. Therefore Gumbel distribution can be considered to fit the discharges of the study catchments well enough.

The annual maximum discharges of the hydrological year from August to September were used for flood analysis. Hydrological year was used to avoid picking the same flood event more than once in the long lasting winter floods, which become more common in some catchments in the future.

The control 100 year flood was defined as follows: the 100 year flood estimated with Gumbel distribution from annual maximum discharges of 1971–2000 simulated with the WSFS hydrological model using observed temperatures and
precipitations. The same floods are the control period floods in the delta change method. The 95 % confidence limits of the control 100 year flood were defined as 1.96 * standard deviation and this range was called the adequate range. If the 100 year floods estimated from the discharges simulated with RCM data, for the control period, was within this adequate range it was found to be an adequate simulation. The performance of the hydrological model was tested by the same method, by comparing the simulated control 100 year flood to the 95 % confidence limits of the 100 year flood estimated from discharge
observations of 1971–2000.

## 3      Results

### 3.1      Precipitation extremes in control period

Without bias correction the RCMs overestimate the annual areal precipitation in the study catchments by on average 28 % compared to values based on observations (Table 2). The 100 year precipitations (daily precipitation with 100 year return
period estimated with GEV distribution) are overestimated on average by 24 % (range 0–106 %) and the overestimation is similar for average values. However, the extreme 5 and 15 days precipitation sums are only 1–2 % larger than corresponding values from observations (range -22– 44 % with different RCMs and different study catchments). The largest overestimation in 1 and 5 day extreme precipitation was with HadRM scenarios and the smallest values were produced by REMO scenario, which underestimates extreme precipitation especially in Ounasjoki (not shown).

The adjustment of the precipitation distribution becomes challenging when RCMs tend to overestimate the average precipitation significantly, but the longer extreme precipitation sums important for flood generation are closer to observations. After DBS adjustment the average precipitation was very close to the observed values (average difference 1 %; Table 2). The daily 100 year areal precipitations are overestimated on average by 8 % (range -46–+78 %) with single gamma method and underestimated by 8 % (range -42–+11 %) with double gamma method. The double gamma method on average
underestimates the 5 and 15 days precipitation sums, which are on average 23 % too small in the northernmost catchment Ounasjoki and 8–11 % too small in the other case study catchments. The single gamma method underestimates the 5 and 15



day extreme precipitation sums systematically only in Ounasjoki (Table 2). In conclusion both DBS adjustments improve the average and daily extreme precipitations compared to uncorrected RCM data, but the double gamma tends to underestimate the 5–15 days extreme precipitation sums.

## 3.2 Precipitation changes in the future

Based on the uncorrected RCM scenarios, the annual precipitation sums of the study catchments increase 0 –27 % (on average 14 %) by 2070-2099 compared to 1971-2000 (Fig. 2). The relative increase in seasonal mean precipitation is largest in winter while the absolute increase is largest in summer. The increases are largest in the northernmost catchment Ounasjoki, but the differences between changes in different RCM scenarios are large (Figs. 2b and 2c). The changes in mean precipitation are smallest in HIRHAM-A scenario and largest in HIRHAM-B, which highlights the importance of the GCM
used to drive the RCM.

Estimated changes in extreme areal precipitation (Fig. 2) show a larger change in the 100 year precipitations than in the average precipitation in the catchments of Loimijoki and Nilakka, but in larger catchments of Saimaa and Ounasjoki the changes in extreme precipitation are closer to changes in annual precipitation. The difference in seasonal extreme and mean precipitation changes are largest during summer, when the average increase is relatively small (on average 12 %), but the
change in 100 year precipitation is the largest (on average 32 % with uncorrected scenarios, Fig. 2a). However, this effect varies between scenarios as HIRHAM-A and REMO produce much larger increases in extreme precipitations compared to average precipitation, whereas in HadRM and HIRHAM-B the increase in extreme precipitation is close to or even smaller than the average precipitation (Fig. 2c).

The DBS adjustment preserves well the changes in mean precipitation with similar changes in the uncorrected RCMs and
DBS adjusted precipitations in all seasons, catchments and RCMs (Table 2, Fig. 2). However, the estimated changes in 100 year precipitations are with some scenarios different with the uncorrected precipitations than with single and double gamma adjusted precipitations. With double gamma correction the 100 year precipitation increases significantly more than with uncorrected RCM and single gamma in the HIRHAM-A and HIRHAM-B and RCA scenarios (Fig. 2c). This behaviour is also seen in the other panels (a, b, d) of Figure 2. Trends of daily maximum precipitation are overall preserved well in DBS
adjustment compared to uncorrected trends. Nevertheless, DBS adjustment with double gamma increases the RCM indicated trend of daily maximum precipitation slightly more than with single gamma.

The changes in extreme precipitations are important for flood generation especially in small watersheds, where severe floods can be caused by heavy summer precipitations. In large and medium sized catchments the summer and autumn floods are typically generated by 5–15 days precipitation sums. In these longer precipitation sums the changes in 100 year
precipitations are closer to the changes in average precipitation than in 100 year 1 day precipitation (Fig. 2d). Thus the tendency of the DBS double gamma adjustment to produce larger changes in daily 100 year precipitation does not necessarily affect the 100 year flood discharges in the study catchments.



### 3.3  Floods in control period

The 100 year floods (daily discharge with 100 year return period estimated with Gumbel distribution from simulated discharges) were estimated from the uncorrected and DBS adjusted RCM data for control period 1971–2000. The uncorrected 100 year flood magnitudes differ remarkably from the observed and control 100 year flood magnitudes. In the

5 case study catchments the average overestimation of the 100 year flood was 65 %, and only 5 % of the simulated 100 year floods of the case study catchments were within the adequate range compared to the control flood (Table 3). For the national data set, on average 14 % (0–53 % in different scenarios) of the 100 year floods from uncorrected RCM data were within the adequate range. The average overestimation of the 100 year flood magnitude compared with the control 100 year flood was 22–95 %. In most scenarios the main reason for the inadequate simulation of flood magnitudes with the uncorrected RCM

data were the overestimation of precipitation. In many scenarios the flood estimates also differed because of the cold bias in winter and spring, which caused large overestimation of accumulated snow, and hence overestimation of spring floods and delay in timing of the spring floods.

The DBS adjustment of temperature and precipitation improved the simulation of floods considerably and made the timing of the floods more consistent with observations. On the four case study catchments the single and double gamma DBS

adjusted 100 year floods were adequate with all five scenarios for Lake Saimaa and Lake Nilakka, three scenarios for Ounasjoki and two scenarios for Loimijoki (on average 75 % of scenarios within the adequate range) (Fig. 3). Despite the significant improvement in flood simulation with DBS method, marked differences between simulated and observed floods still remain in some scenarios (Figs. 3 and 4). Even though the average precipitation sum and average temperature are corrected relatively well, the biases in above zero temperatures and remaining biases in precipitation cause too much snow

accumulation and too large spring floods in some scenarios, e.g. REMO scenario in Loimijoki (Fig. 4).

The number of adequate scenarios in the national data set increased to 60 % (22–81 % in different scenarios) with single gamma and 62 % (33–83 %) with double gamma DBS. The number of adequate scenarios for each site was on average 3 with single gamma method (Fig. 5), but ranged from 1 (two sites) to 5 (10 sites). The worst performing RCM by this criteria was the REMO model with only 22 % of the study catchments accepted with single gamma DBS and 33 % with double

gamma DBS. The number of adequate scenarios was highest in central Finland lake area, while in the small southern rivers the number of adequate scenarios was low (Fig. 5), because of the climatological and hydrological properties of the catchments. The southern rivers are most sensitive to daily precipitation and therefore more sensitive to the remaining biases of the precipitation time series. In addition the winter temperature of the southern rivers is often close to zero in the control period and therefore biases in winter temperature can influence the snow accumulation and melt. Even though the RCM have

rather good resolution, the temperature and precipitation gradient in the coastal areas may cause biases especially in the small coastal rivers.

The differences between 100 year floods simulated with single and double gamma adjusted RCM data were in general small. With the double gamma adjustment the 100 year flood differed from the control flood by -18–33 % and with single gamma



adjustment by -15–35 %. With double gamma adjustment a slightly larger number of catchments of the national data set were within adequate range than with single gamma adjustment.

In addition to adequate bias correction, the performance of the hydrological models should be critically assessed. In Figure 6 square symbols show the locations of inadequate performance of the hydrological model compared to observed discharges,

5    which occur in total on 6 out of 67 sites in the national data sets. The criteria for adequate model performance was similar to testing the DBS method, i.e. the 95 % range of the 100 year flood estimates based on the observed discharges was compared to 100 year flood estimates of the hydrological model simulation with observed temperature and precipitation. The inadequate performance of the model is mostly caused by the lake regulation, which affects the 100 year flood estimates, but in some cases the hydrological model performance or input data is just not valid for the extreme flood analysis.

### 3.4    Floods in the future

The changes in floods were estimated by comparing the 100 year floods estimated from simulated discharges of 2070–2099 from each RCM scenario to corresponding simulated flood estimates of 1971–2000. In the case study catchments the results vary from consistent decreases in 100 year floods in Ounasjoki to consistent increases in Lake Saimaa (Table 4). In the

northern Ounasjoki the floods in the control period are caused by snowmelt in spring and all the scenarios and methods project decreasing floods by 2070–99 due to decreasing snow amounts. The catchment of Lake Saimaa is large and has a high lake percentage causing long delays to the precipitation events. The floods are generated by long-term (1–5 months) precipitation sums and increase due to increasing precipitation and milder winters. In Lake Nilakka and Loimijoki different RCM scenarios produce larger variation of projected changes in 100 year floods with a range from 30 % decrease to 25 %

increase by 2070–99.

The results from the national data set for 2070–2099 (Fig. 5) with single gamma show that on most catchments the annual 100 year floods are projected to decrease. The largest decreases are with REMO scenario, which also has the largest number of inadequately performing catchments. Largest 2070–2099 floods are produced with HIRHAM-B scenario, which also had the largest increases in precipitation. Increases are mainly projected the lake area in central Finland, especially for the two

largest catchments (Lake Saimaa and Kymijoki). The range of changes projected is large with at least one scenario projecting increasing floods (more than 10 % increase from control period flood) on 36 % of the study catchments (Fig. 5).

The changes in 100 year floods in the national data set with double gamma (not shown) were very similar than with single gamma (Fig. 5). On average the difference in projected changes between single gamma and double gamma was 0.9 % with single gamma producing slightly larger values on average.

The 100 year floods were also simulated with delta change method to enable comparison of projected changes in floods with the two methods (Table 4, Figs. 5 and 6). In the case study catchments the delta change method produces on average larger 100 year floods for 2070–2099 in three catchments (Lake Saimaa, Loimijoki, Ounasjoki) (Table 4). In Lake Nilakka the delta-change method produces a narrower range (-1–5 %) of change with different RCMs than single and double gamma

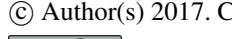



methods, -26–26 % and -29–20 % respectively. In Loimijoki the delta change method also produces a smaller range of changes than the DBS adjustment, but with three scenarios the DBS adjustment does not produce the control period floods adequately well. In addition to these three scenarios in Loimijoki, two scenarios in Ounasjoki fail to produce 100 year floods in adequate range in the control period. Leaving out scenarios that performed poorly would affect the average result

especially in Loimijoki. In Loimijoki the scenarios not performing adequately produced too large spring floods and too much snow in the control period resulting in large reductions in floods in 2070–2099 (Fig. 3).

With catchments of the national data set the delta change method produces on average 1.4 % larger changes in floods in 2070–99 period than the single gamma DBS adjusted simulations (Figs. 5 and 6) and 2.3 % larger than double gamma. The largest differences between the methods were in south and south-eastern Finland, where floods may increase or decrease

depending on the climate scenario and downscaling method. In this region the floods are sensitive to the changes in extreme precipitation and changes in winter temperature. In small coastal rivers flowing to the Bothnian Bay delta change method produced significantly smaller decreases in floods than the DBS adjusted RCM data (Fig. 5 and 6). By contrast some catchments in eastern Finland and north central Finland also produced larger increases with the delta change approach. The range of changes from the five scenarios is considerably larger with DBS method than with delta change. This is partly

caused by the remaining biases in the control period, but also due to natural variation, which affects the results produced from the 150 years RCM simulations, while the temporal variation of delta-change method is dependent on observed weather in the control period.

The influence of the scenarios not performing adequately was on average relatively small, but in some catchments these scenarios can distort the results. This is evident especially in the smaller rivers in the south, where poorly functioning RCM

scenarios produced too large spring floods in the control period (similar to Loimijoki, Fig. 3, Table 3) resulting in unrealistically large decreases in 100 year floods in 2070–2099 (especially scenarios REMO and RCA, which use ECHAM5 global model). In these catchments the adequately functioning scenarios produce much smaller decreases in floods or even increases. Therefore it is recommended to reject the inadequately performing RCM scenarios so that the results become more reliable.

**4        Discussion**

Both versions of the DBS bias correction method, single and double gamma, which were compared in this study, improve significantly the RCM daily and monthly precipitation sums compared to observed precipitation sums. Yang et al. (2010) found that the use of double gamma distribution improved the reproduction of the full range of precipitation distribution compared to observations for one RCM. In our results the single and double gamma DBS methods produce distinct

differences at the high end of the precipitation distributions. The double gamma distribution decreases heavy precipitations and the magnitude of estimated 100 year precipitation of the control period too much compared to observations with most RCMs. The main reason for the difference is the uncertainties in the tail of the distribution. Double gamma method divides





the precipitation distribution into two parts, below and above 95% of the cumulative distribution. This separation made for every month leaves the extreme 5% scant and makes it sensitive to few observed occurrences of heavy precipitation, which are relatively rare in the Finnish climate. When this is combined with the sensitivity of the GEV parameters estimated for each month and grid cell separately and the changes in precipitation in the future time periods, the results can sometimes

become questionable. As a consequence, in some RCMs the percentage increase in double gamma 100 year precipitation by 2070–2099 was considerably larger than with single gamma or with uncorrected values, which can potentially cause an unconfirmed signal in extreme precipitation and in floods caused by extreme precipitation. Due to the inconsistencies of the double gamma compared to uncorrected RCM results in estimated changes of 100 year precipitation, the single gamma seems more robust and more ready for use in flood studies.

The differences between single and double gamma in simulated floods were small. Bias correction of precipitation using single gamma distribution has been proven effective in several studies (Block et al., 2009; Piani et al., 2010; Teutschbein and Seibert, 2012). This study also shows that the single gamma DBS adjustment produces sufficient correction in most cases even when extreme floods are examined. Before the use of double gamma distribution can be recommended in Finland, further improvements would be required. For example different separation point for the two distributions of the double

gamma method could be tested. Yang et al. (2010) suggested a flexible threshold to allow a better fit under varying circumstances. Also longer time window (e.g. 90 day window used to estimate parameters for each month) or clustering grid cells for parameter estimation could produce more stable parameters for the two distributions of the double gamma method.

The flash floods and floods in small catchments can be caused by one day or even shorter heavy precipitation events. In these cases the changes in extreme precipitation and the correct simulation of daily extreme precipitation is important.

However, the extreme precipitation sums of several days (5–15 or even longer) are usually the driving factors of the summer and autumn floods in larger catchments in Finland. 100 year return periods for 5 and 15 days precipitation sums were underestimated with double gamma DBS adjustment and with single gamma in Northern Finland in Ounasjoki. The DBS adjustment lead to worse estimates for extreme 5 and 15 days precipitation in those cases, where the gamma distribution did not fit well to the observations. Addor and Seibert (2014) also found that while a non-parametric quantile mapping bias

correction was effective in reducing daily biases, large biases still remained in the multiday to interannual precipitation characteristics in most stations. Gudmundsson et al. (2012) showed that non-parametric transformations correct the biases more efficiently than gamma distribution. On the other hand use of non-parametric empirical distribution has challenges to deal with the end tail of the distribution and especially with "new extremes" (Maraun et al., 2012; Chen et al., 2013). Therefore the use of several bias correction methods is recommended (Chen et al., 2013; Räisänen and Räty, 2012; Räty et

al., 2014) and careful validation should be carried out to take into account the uncertainties of the methods used.

The correction of precipitation and temperature with DBS methods improved the simulation of discharges and floods considerably. However, despite the improvement with the DBS adjustment, some RCM scenarios were still not able to produce the observed control period floods adequately due to remaining biases in RCM precipitation and temperature, the performance of the hydrological model, the performance of the flood frequency analysis and/or natural variation in the




control period. In some cases the remaining biases after the bias correction influence the climate change signal produced by the simulations. In our results this occurs most commonly because control period spring floods are larger than observed due to too large snow amount, which leads to very large decreases in floods in future periods. The quality of the discharges simulated with the RCM scenarios should be critically assessed, before the results are presented and used in impact studies.

The scenarios that are unable to match the control period hydrology should be identified and possibly rejected.

In this study we compared the 100 year floods from the RCM simulation to the 95 % uncertainty range estimated for the control 100 year floods (floods simulated with observed temperature and precipitation) to assess the performance of the simulations. This method is used for testing the adequate performance of the DBS adjusted RCM scenarios and hydrological model performance in producing the annual maximum discharges. For more comprehensive testing a similar or improved

method could also be applied for different seasons, because the climate change affects the timing and the driving factors (snowmelt or heavy rain events) of the annual maximum discharges. While most scenarios performed relatively well after the DBS correction, especially the REMO scenario was not able to simulate the control period floods adequately in most of the catchments. The entire scenario should therefore be evaluated carefully and the results should not be used for further impact studies. The ECHAM5 model used as boundary condition for the REMO has problems simulating air temperatures

especially during snow melt period due to e.g. inadequacies in surface temperature simulation when the ground is partially covered in snow (Räisänen et al., 2014). The biases in near zero temperatures remain even after the temperature is bias corrected (Olsson et al., 2015). Maraun (2013) concluded that in cases when the bias is caused by incorrect responses of the RCM processes, such as surface albedo and soil moisture, the bias correction can even deteriorate future simulations and increase future bias.

The five DBS adjusted scenarios produced a very large range of climate change signal by 2070–2099. A large range of results can in some cases be realistic, because of different changes in temperature and precipitation in the different climate scenarios and natural variability. However, in other cases the range produced can be unrealistically large due to unrealistic control period floods simulated with the RCM data. The unrealistically large range of results may lead to disregarding of the climate change effects, if the presented uncertainty of the results hides the true climate change signal, or the large uncertainty

can cause too large adaptation measure to be demanded if the worst case scenarios are used as basis of planning.

Besides the changes in flood magnitudes, also the timing of the floods and the relative magnitudes of floods in different seasons are important. The biases in the RCM results can vary in different seasons, due to different driving factors in winter, spring, summer and autumn floods. In winter and spring most of the floods are caused by snowmelt and the biases in winter precipitation sums and temperatures are the main reasons for biased flood magnitudes. The summer and autumn floods are

produced by the heavy precipitation events lasting from couple of days to couple of weeks in larger catchments. The model ability to correctly simulate magnitudes of floods in different seasons is important in the climate change studies, because floods in different seasons change differently in response to climate change (Veijalainen et al., 2010). In a country such as Finland, snow accumulation and snowmelt are key factors affecting the winter and spring floods. Therefore the bias





correction method used for temperature correction can be as important as the method for precipitation correction for obtaining realistic discharges and floods.

Comparison with results from Veijalainen et al. (2010), where 20 climate scenarios and delta change method (same 67 catchments as in the national data set) was used with Gumbel distribution to estimate changes in 100 year floods by 2071–2100, shows that the results are on average similar. However, the range of projected changes in floods with DBS adjustment and five RCM scenarios is even larger than with delta change and 20 GCM and RCM scenarios. This is partly explained by the inclusion of natural variability in the DBS method and partly by the unrealistic results of some scenarios mentioned earlier. When using only the scenarios with adequate control period flood simulation, the range of results is similar as with 20 scenarios of delta change method, but the number of scenarios used in DBS method decreases from 5 to on average 3 per catchment. The results between delta change and DBS adjustment may differ because DBS method takes into account the effect of climate change on temperature and precipitation variability and because of remaining biases in DBS adjustment, different properties of the control period and natural variability (Graham et al., 2007; Beldring et al., 2008)

The modelling chain of the hydrological climate change impacts is long and complex and includes uncertainties in every step. The sources of uncertainties include emission scenarios (not considered in this study, only A1B), global climate models (four different used), regional climate models (four different), bias correction methods (two versions of one method for precipitation, single and double gamma), hydrological model and its parameterization (not considered, one model and parameter set) and flood frequency analysis (GEV distribution for precipitation and Gumbel for discharge). Several studies have estimated the relative importance of different sources of uncertainty (Steele-Dunne et al., 2008; Kay et al., 2009; Prudhomme and Davies, 2009; Vormoor et al. 2015, Karlsson et al., 2016). These studies have found that the largest source of uncertainty is usually the global climate model, but other sources can be important as well. Räisänen and Räty (2013) concluded that while generally the climate model simulations are associated with larger uncertainty than the choice of bias correction method in temperature, the relative importance of the method increases towards the tail of the distribution i.e. when extremes are in question. Teutschbein and Seibert (2012) and Räisänen and Räty (2012) recommended using several different well performing bias corrections to account for this uncertainty. However, our results show that in impact assessment the performance of the simulations should be carefully evaluated to avoid unrealistic results due to remaining biases, and thus unnecessarily adding to the already large uncertainty range.

## 5 Conclusions

Two versions of the DBS bias correction method, single and double gamma distribution, were used in this study to correct RCM simulated precipitations. Both distributions produce similar monthly mean values. According to the results, extreme summer precipitations increase more than the average precipitation by the end of this century, while in other seasons no significant differences between extreme and average precipitations were found. The double gamma distribution produces larger changes in 100 year extreme precipitations by 2070–2099 than the corresponding changes calculated from uncorrected



RCM data, and the single gamma distribution. In Finnish conditions with limited amount of heavy precipitation events, the single gamma distribution is probably more suitable when extremes are studied.

The bias correction of RCM data is necessary when hydrological impact assessment is carried out, since without it the simulated floods differed significantly from observations due to biases in RCM temperature and precipitation. Also the simulated changes in floods would have been significantly different since the biases in temperature and precipitation affect which types of floods are simulated. E.g. snowmelt and heavy rain induced floods have different responses to climate change. However, the type of the distribution of precipitation correction (single or double gamma DBS adjustment) had only minor effect to the projected changes in floods, because in most of the catchments in Finland the largest floods are caused by snowmelt, or 5-15 days extreme precipitation sums, rather than one day extreme precipitation events.

Effects of climate change on floods in Finland vary regionally due to differences in climate and hydrological properties of the catchments. Spring floods generated by snowmelt mainly decrease, while autumn and winter floods tend to increase. The results are in general similar as with delta change method, but the DBS adjusted RCM data produces a larger range of results. Climate change impacts on snowmelt floods and floods in large catchment, which depend most on average changes in precipitation and temperature, can be estimated relatively well with all single and double gamma and delta change methods. The floods caused by short-term extreme precipitation can be affected more by the method used.

The range of results with five DBS adjusted RCM scenarios is large. This is due to differences in future scenarios and in natural variability, but also because of remaining biases in RCM temperature and precipitation data and the unrealistic results these may cause. If the control period floods and their flood generating processes are simulated unrealistically, the estimated future changes in floods may also be unrealistic. The estimation of the performance of scenarios is therefore needed, and in some cases poorly working scenarios need to be rejected to avoid misleading results. A method used for estimating the model performance and acceptability of the simulations based on normal uncertainty range is applied in this study, and could be further developed for achieving more reliable climate change impact assessments.

**Acknowledgements**

This study was carried out part of the project Climate Change and Water Cycle: Effect to Water Resources and their Utilization in Finland (ClimWater) (no. 140930) financed by the Academy of Finland as part of the Research Programme for climate change FICCA and as part of ELASTINEN-project financed by VNK. Authors also received support from Strategic Research Council funded Winland project. The ENSEMBLES data used in this work was funded by the EU FP6 Integrated Project ENSEMBLES (Contract number 505539) whose support is gratefully acknowledged. We would also like to thank Kalle Sippel for his work on drawing the maps for this paper





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



**Table 1. Regional climate model (RCM) data used in this study. Data is from ENSEMBLES data base (ensemblesrt3.dmi.dk; van der Linden and Mitchell, 2009)**

| Name/Acronym | RCM | GCM | Emission scenario |
| --- | --- | --- | --- |
| HIRHAM-A | HIRHAM5 | ARPEGE | A1B |
| HIRHAM-B | HIRHAM5 | BCM | A1B |
| REMO | REMO | ECAHM5 | A1B |
| RCA | RCA | ECHAM5 | A1B |
| HadRM | HadRM3Q0 | HadCM3Q0 | A1B |



**Table 2. Average and 100 year precipitations (1, 5 and 15 day) (mm) in 1971–2000, average of the four study catchments. Precipitations from observations (control) and from RCMs without correction and with single and double gamma DBS adjustment.**

| Scenario | Average | 100 year 1 day | 100 year 5 day | 100 year 15 day |
|---|---|---|---|---|
| **Control (mm)** | | | | |
| Loimijoki | 1.74 | 32.0 | 78.4 | 143 |
| Lake Nilakka | 1.81 | 45.4 | 71.6 | 116 |
| Ounasjoki | 1.52 | 49.0 | 91.8 | 128 |
| Lake Saimaa | 1.91 | 29.4 | 101.0 | 146 |
| **Uncorrected RCM precipitations (% compared to control)** | | | | |
| Loimijoki | 7…+30 % | 28…+106 % | 1…+44 | -22…+18 |
| Lake Nilakka | 4…+23% | 0…+41 % | -15…+33 | -3…+12 |
| Ounasjoki | 13…+49% | -38…+11 | -22…+19 | -12…+7 |
| Lake Saimaa | 6…+30 % | 33…+101 | -17…+32 | -5…+23 |
| **DBS corrected RCM precipitations (single gamma) (% compared to control)** | | | | |
| Loimijoki | 2…+4% | 19…+78 | -0…+35 | -19…+10 |
| Lake Nilakka | -1…+1% | -6…+21 | -29…+18 | -8…+5 |
| Ounasjoki | 0…+3% | -46…-21 | -44…-10 | -38…-1 |
| Lake Saimaa | -1…+2% | -4…+62 | -20…+19 | -0…+8 |
| **DBS corrected RCM precipitations (double gamma) (% compared to control)** | | | | |
| Loimijoki | 2…+5% | 6…+11 | -8…+11 | -24…-10 |
| Lake Nilakka | -1…+2% | -18…+19 | -30…+8 | -16…+2 |
| Ounasjoki | 0…+3% | -42…-10 | -43…-16 | -36…+0 |
| Lake Saimaa | -1…+2% | -17…+12 | -28…-6 | -11…+0 |



**Table 3. Estimated 100 year floods (m³s⁻¹) of the case study catchments in 1971–2000. Floods estimated from observations, from simulated with observed T and P and from simulated with uncorrected RCM data and with single and double gamma DBS adjusted RCM data.**

| Scenario | Loimijoki | Lake Nilakka | Ounasjoki | Lake Saimaa |
|---|---|---|---|---|
| Obs | 335 | 76.4 | 1670 | 1320 |
| Control | 325 | 71.2 | 1566 | 1080 |
| Uncorrected RCM data | | | | |
| HIRHAM-A | 590* | 130* | 3090* | 1490* |
| REMO | 603* | 117* | 3120* | 1590* |
| RCA | 552* | 75.3 | 1950* | 1560* |
| HadRM | 575* | 93.5* | 2170* | 1480* |
| HIRHAM-B | 656* | 115* | 3000* | 1480* |
| DBS adjusted RCM data (single gamma) | | | | |
| HIRHAM-A | 406* | 67.5 | 1660 | 1090 |
| REMO | 523* | 81.4 | 2310* | 1230 |
| RCA | 515* | 73.3 | 2030* | 1080 |
| HadRM | 323 | 56.9 | 1610 | 1130 |
| HIRHAM-B | 307 | 63.5 | 1500 | 1190 |
| DBS adjusted RCM data (double gamma) | | | | |
| HIRHAM-A | 405* | 67.7 | 1660 | 1090 |
| REMO | 514* | 80.1 | 2270* | 1200 |
| RCA | 506* | 72.9 | 2030* | 1070 |
| HadRM | 315 | 57.0 | 1620 | 1120 |
| HIRHAM-B | 297 | 63.2 | 1490 | 1180 |

*not adequate simulation of control 100 year floods



**Table 4. Comparison of changes in 100 year floods (m³s⁻¹) by 2070–2099 with delta change and with DBS adjusted (single and double gamma) RCM data in case study catchments with different climate scenarios. Note that the value of the control 100 year flood is different in each RCM scenario and in delta change (same for all the scenarios with delta change, simulated with observed T and P).**

| Scenario | Loimijoki | Lake Nilakka | Ounasjoki | Lake Saimaa |
|---|---|---|---|---|
| DBS adjusted RCM data (single gamma) % | | | | |
| HIRHAM-A | -38.6* | -10.5 | -32.8 | 19.7 |
| REMO | -47.1* | -25.9 | -37.0* | 10.1 |
| RCA | -50.5* | -11.6 | -40.7* | 27.6 |
| HadRM | -11.2 | 14.9 | -44.2 | 19.3 |
| HIRHAM-B | -3.0 | 26.0 | -21.0 | 25.9 |
| DBS adjusted RCM data (double gamma) % | | | | |
| HIRHAM-A | -41.5* | -11.4 | -34.3 | 18.8 |
| REMO | -47.0* | -28.7 | -36.4* | 8.7 |
| RCA | -51.6* | -10.6 | -40.0* | 28.0 |
| HadRM | -21.3 | 15.1 | -42.3 | 22.7 |
| HIRHAM-B | 1.2 | 19.5 | -25.3 | 24.0 |
| Delta change | | | | |
| HIRHAM-A | -27.8 | -10.2 | -20.2 | 10.8 |
| REMO | -19.8 | -7.3 | -15.5 | 21.5 |
| RCA | -18.1 | 5.1 | -8.6 | 34.8 |
| HadRM | -13.0 | -3.7 | -17.7 | 17.9 |
| HIRHAM-B | -15.2 | -1.2 | -38.4 | 29.7 |

5  *not adequate simulation of control 100 year floods





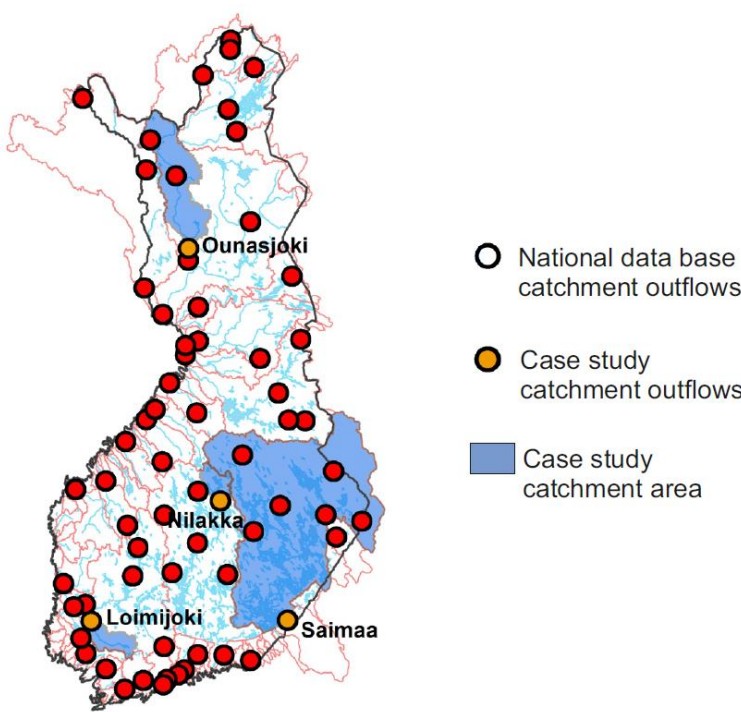

**Figure 1. Map of study catchments. Circles (orange and red) are catchment (discharge station locations) of the national data set, orange circles are also case study catchments (discharge station locations) and blue areas are the upstream catchment areas of the case study catchments.**



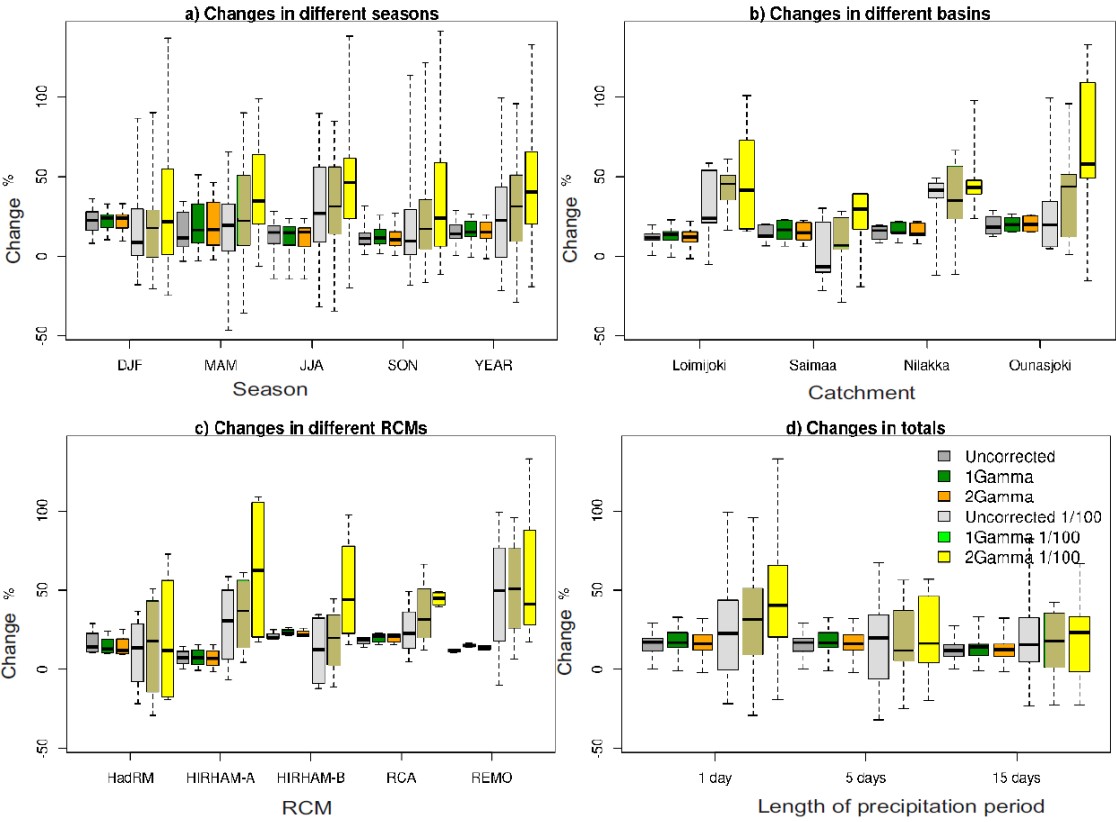

**Figure 2. Changes (%) in mean and extreme (100 year return period) precipitation in different a) seasons, b) catchments, c) RCMs and d) precipitation sums in 2070–2099 compared to 1971–2000. Figures represent (a) seasonal and annual average values of five RCMs and four catchments, (b) annual average values of five RCMs, (c) annual average values of four catchments, and (d) annual average values of 1/5/15 day precipitation sums of five RCMs and four catchments.**





**Figure 3. Simulated maximum daily discharges (m³s⁻¹) of the hydrological year (Sept-Aug) and the Gumbel distribution fitted to the maximum discharges in the case study catchments in control period 1971-2000 a) Ounasjoki, b) Lake Nilakka, c) Loimijoki, d) Lake Saimaa. Results are simulated with single gamma (1Gamma) DBS adjusted RCM data and simulated with observed T and P.**





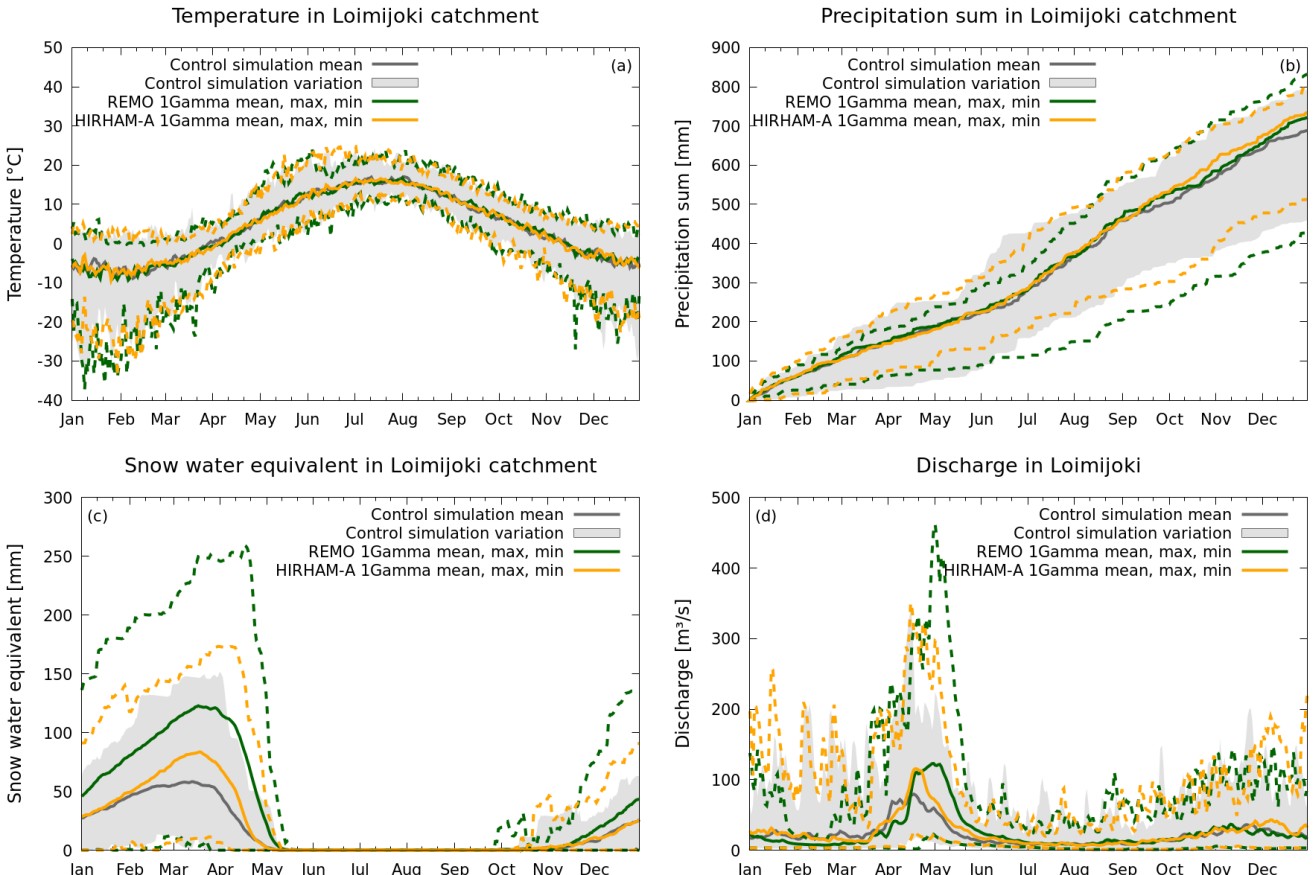

**Figure 4. Maximum, average and minimum daily a) temperature (°C), b) precipitation sum (from 1st of Jan) (mm), c) snow water equivalent (mm), and d) discharge (m³s⁻¹) for control simulation (model simulated with observed T and P) and for single gamma (1Gamma) DBS with REMO and HIRHAM-A RCM scenarios.**





**Figure 5. Changes (%) in floods in the catchments of the national data set with DBS single gamma correction in 2070–2099 compared to 1971–20000 with a) HIRHAM-A, b) HIRHAM-B, c) REMO, d) RCA and, e) HadRM RCM scenarios. Also the function of the simulation of floods in the control period with RCM data compared to control floods is shown.**






**Figure 6. Changes (%) in 100 year floods with delta change in the catchments of the national data set in 2070-2099 compared to 1971–2000 with a) HIRHAM-A, b) HIRHAM-B, c) REMO, d) RCA and, e) HadRM RCM scenarios. Also the function of the simulation of floods in the control period with the hydrological model compared to observed floods is shown.**

