# Peer review of "Impacts of climate change on extreme floods in Finland –studies using bias corrected Regional Climate Model data"

_Hydrology and Earth System Sciences, 2017_

## Referee Comment (RC1) · Anonymous Referee #1 · 4 Jan 2018

Comments on: Impacts of climate change on extreme floods in Finland – studies using bias corrected Regional Climate Model data. Veijalainen, N., Jakkila, J., Olsson, T., Backman, L., Vehvilainen, B., Kaurola, J. Submitted for consideration for publication in Hydrol. Earth Syst. Sci.

General statement: The manuscript presents projected changes in the 100-year precipitation and the 100-year discharge for a set of 67 catchments in Finland, including detailed results for four catchments and a general overview of the results for the full set of catchments. The projected changes are based on precipitation and temperature series from 5 RCM runs from the ENSEMBLES project. These series are bias corrected using both single gamma and double gamma distribution-based correction before further analysis, and the results are then compared. The results are also compared with hydrological projections derived using change factors applied directly to observed timeseries. The authors conclude that the differences between the projected floods estimated with precipitation adjusted using a single vs. a double gamma distribution-based correction are small, and in addition, the results based on bias-corrected RCM are generally similar to those estimated using change factors.

The assessment of likely changes in flooding under a future climate using a systematic approach, such as described in this work, is clearly of national importance for Finland. Comparison of the performance of different RCMs in producing floods of the correct magnitude during the control period is also a noteworthy component of this procedure. However, I fail to see that this manuscript, in its present state, presents methods or results which can be considered to be both new and of international relevance. It seems that the authors are aware of the need to highlight important methodological issues and have chosen to focus on a comparison of bias adjustment using single vs. double gamma functions. Simultaneously, the four case study catchments they present are all relatively large (> 2160 km2), two of them have a high lake percentage (18-20%) and at least one of them is significantly affected by regulation. In these cases, it is very unlikely that two methods which differ only in the correction of the highest quantiles (i.e. > 95%) of DAILY precipitation will give systematically different results which can be distinguished in the flood quantiles. The authors briefly acknowledge this in their discussion in that they suggest that it is the 5-15 day precipitation sums that are important for driving floods (outside of the snowmelt season) in many Finnish catchments. However, this raises the question as to why these two bias correction methods, which only differ in their adjustment of the highest quantiles, are being compared in these catchments (i.e. catchments with relatively long response times). It may well be that some of the 67 catchments in the national data set are small enough to have shorter response times such that the differences between the two methods could be discerned. No information on the catchment characteristics of the large data set is, however, provided. The authors also seem to be unaware of several recent studies which have compared bias correction and other adjustment methods (e.g. delta change and, more generally, perturbation methods) for extreme precipitation (e.g. Sunyer, et al., 2015, HESS; see also Willems and Vrac, 2011, J. Hydrol. ) and their effect on projections of extreme discharge (e.g. Hundecha, et al., 2016, J. Hydrol., Osuch, et al., SERRA, 2016).

In summary, I suggest that the authors need to reformulate their objectives and focus before this study can be deemed suitable for publication in HESS and of interest to an international audience. A number of specific suggestions in this direction are given below:

Specific comments:

1. The 'delta change' method is presented as a conventional method used in standard practice and that the use of bias adjusted daily data from RCMs have 'recently become more common' (pg. 2 l. 17). This statement may well have been true 10 years ago, but climate impacts research is indeed a fast-moving research are. There are innumerable examples of the use of daily bias-adjusted RCM data for hydrological impacts research available today in the literature. I suggest that this section of the manuscript could be improved significantly by including a more comprehensive review of current work considering the effect of bias correction on extreme precipitation and projected changes in flooding. Some of the relevant work is already mentioned in the discussion (e.g. Gudmundsson, et al., 2012; Beldring, et al., 2007) but should come earlier in the manuscript. Other sources are given above in the general comments, others can be found in Madsen, et al., J. Hydrol., 2014, or indeed by a more comprehensive literature search than is evident in this manuscript.

2. The preservation of trends is an important point to consider when using bias correction, and this is also mentioned here. It has, however, been suggested by Hempel, et al. (Earth Sys. Dynamics, 2013) that a 'trend-preserving bias correction' in which

residuals are corrected after removing the trend, generally gives better results. Has this been considered or tested? I realise that a full implementation of this approach would require that the entire work be redone, but the implications of this could at least be tested for precipitation.

3. The spatial resolution of the ENSEMBLES RCM grids is not mentioned at all in this manuscript, although it is clearly relevant for the ability of an RCM to reproduce spatial and temporal patterns of extreme precipitation. The most recent publications presenting changes in hydrology under a future climate are based on EUROCORDEX RCM runs, including runs with grids having a higher spatial resolution than ENSEMBLES runs.

4. The focus of the extreme value analysis here is the assessment of the 100-year flood as it is of national importance for climate change adaptation. Given the uncertainties introduced by fitting an extreme value function and the short annual maximum series available for each time slice, it would be more reasonable to first assess changes in the mean annual maximum precipitation and mean annual maximum flood and/or changes in average over threshold values, before presenting results for longer return periods.

5. The use of a Gumbel distribution for modelling extreme discharge values is justified by the application of the likelihood ratio test. Was this test applied to all of the data series for each catchment (i.e. including both control and future periods modelled from the corrected RCM data) or has this only been assessed based on observed time series? The use of a Gumbel distribution as a basis for assessing changes also raises the question as to whether or not changes in the tails of the distribution (i.e. the most extreme values) can be fully detected by such an approach. This is a particularly important point here in that changes in extreme precipitation are assessed based on a comparison of GEV fits (i.e. with a shape parameter) and for discharge with a Gumbel distribution (i.e. without a shape parameter).

Other minor technical comments

1. The use of English is generally very good, although there are numerous small errors, particularly related to the use of indefinite and definite articles.

2. The title of the article is too general and does not highlight an important methodological or other contribution of this work.

3. As mentioned above, additional information regarding the characteristics of the 67 catchments representing the national dataset should be provided. This can be given as, for example, mean, minimum and maximum values of variables such as catchment area, lake percentage, degree of regulation (if possible). Some information as to the seasonality of maximum discharge (or ideally the major process driving flooding) would aid significantly in interpreting the spatial distributions shown in figures 5 and 6.

4. Page 11, l. 2-3. 'This separation made for every month leaves the extreme 5% scant and makes it sensitive to few observed occurrences of heavy precipitation, which are relatively rare in the Finnish climate'. The last part of this sentence doesn't entirely make sense, in that it is the upper 5% of the total distribution for each month which is being considered and fitted to a gamma function. For dry months without a large number of rainy days, it could well be the case that there are relatively few values over the 95% threshold, but there are then also proportionally fewer days under the 95% threshold. So, this would not reflect 'few observed occurrences of heavy precipitation', but rather, 'few occurrences of days with precipitation'. It is for this reason that some researchers have chosen to use 3-month blocks for bias correcting precipitation, so that more extreme values are available.

5. The source of the precipitation and temperature data used for bias correction should be discussed, including how areal values are derived for catchment precipitation. Has bias correction been performed relative to station data, or to grid points in a gridded dataset, or to areal values derived from station or gridded data?

6. In Table 1, the column ' Emission Scenario' is superfluous as all are run under the A1B scenario. This could be mentioned in the Table description.

7. The discussion section could benefit from a better structure. It is relatively long, and no overview is given highlighting what topics are covered.

8. A very minor point....but nevertheless important when one is reviewing a paper: The paragraph structure needs to be delineated with, for example, an indentation of the first line, otherwise it can be a bit ambiguous as to when a new paragraph begins.

---

## Referee Comment (RC2) · Anonymous Referee #2 · 5 Jan 2018

The authors employ the DBS method (in two versions) to bias-adjust RCM projections before using them to drive a hydrological model in order to assess the climate change impacts on discharge in Finland. In the DBS method, both a single and a double gamma approach is used, and the results are compared with earlier results btained using Delta Change (DC). The impacts vary within Finland, largely depending on whether floods are generated by snow-melt or rainfall, and the differences between the two DBS versions are overall small.

The topic is interesting and relevant, the material/methods/results are accurate as far as I can judge and the presentation is overall fine. However, I share several of the

concerns raised by reviewer #1, in particular:

- Limited novelty and general significance. Assessing the impact of climate change on extreme floods in Finland is, as I see it, today an engineering excerise rather than scientific research (if "standard methods" are used, as here). And the added value of assessing the impact of "gamma-type" in the DBS method limited, see further next item.

- Doubtful choice of study basins. It is hardly surprising that only very little impact of DBS version on the extreme floods is found, considering the large size of the basins and the associated large-scale nature of the flood generating mechanisms. Smaller/faster basins are probably needed to detect differences related to bias-adjustment of daily P extremes.

- Unclear significance/relevance of the precipitation data used. Essentially no information is given or assessment made about neither the observations nor the RCM data used. About observations, we need to know how they were obtained (interpolated from gauges?), spatial resolution, etc. About RCM data, how does the ENSEMBLES projections relate to more recent, more high-resolution, RCP-based projections (EU-ROCORDEX)? There is nothing wrong with using the ENSEMBLES projections but I think some effort should be spent in putting them in an updated context considering that they are about a decade old.

Having said that, there are interesting parts that could perhaps be further explored, in particular related to the "adequacy issue":

- Rejecting individual projections based on their historical performance is getting more and more accepted, I think. The authors look at how well the hydrological results (extreme floods) based on bias-adjusted historical forcing agree with observations and then identify "adequate simulations". The implications for the final results are however rather briefly discussed and it is a bit hard to see the "big picture". A more systematic analysis covering all projections and basins would be interesting.

[Figure]

- It is further interesting (although perhaps not surprising) that when considering only "adequate simulations", the results approach those obtained by a DC approach. This highlights the trade-off between additional flexibility on one hand and additional complexity/uncertainty on the other hand, in DBS (or, more generally, quantile mapping type of bias adjustment) as compared with DC. This is definitely not the first case when the results of relatively elaborate and complex bias-adjustment applications end up very similar to much simpler DC calculations.

- The adequacy of the hydrological model is very important but only rather briefly discussed. Maybe inadequate models should have been disqualified from the start? Generally, the authors could have expored the "adequacy issue" further, assessed the impact on the results/conclusions in amore systematic way, maybe suggested some kind of general guidelines for how to use this concept.

Specific comments (page/line(s)):

- 6/20: "range 0-106 %", is this correct?

- Table 2: I think you need to include also the mean/median of the RCMs, not only the range.

- 8/8-9: "The average"..."was 22-95 %", difficult to interpret.

- Fig. 4: I have some problems distinguishing between the dark green and dark grey lines, more different colours would help.

---

## Author Comment (AC1) · 15 Feb 2018

Dear Reviewer

We would like to thank you for your useful comments on our manuscript. Here are our replies to the comments.

Choice of study basins is re-evaluated with the addition of Hypöistenkoski to better enable the impacts of the method used on floods caused by daily precipitation to be evaluated on small catchment. Hypöistenkoski is a small river in south-western Finland with 325 km catchment area and 1 % lake percentage. The results of Hypöistenkoski

(see attached Figure) show that there is no significant difference in the performance of the single gamma and double gamma distribution in correcting of the snowmelt floods as expected. However, the single gamma method corrects the maximum annual rain induced floods better than single gamma. Typically the annual maximum floods in Finland have been snowmelt floods but in the future in southern and coastal part of Finland the annual maximum floods will be more frequently rain induced floods due to climate change. Therefore we will add the results of Hypöistenkoski catchment and discussion about the importance of the performance of the bias correction methods for producing snowmelt and rain induced floods separately.

We will also re-evaluate the objectives of the study by focusing more on e.g. exploring the adequacy issue (e.g. guidelines for assessing adequacy) and less on the evaluation of climate change impacts on floods.

The observations are based on Finnish Meteorological Institute gauge observations and as described in Olsson et al. 2015: "The areal values of the meteorological observations are calculated for each sub-basin of the hydrological model from three closest observation stations by inverse distance weighting taking into account the elevation differences. The areal values were converted to the same regular 0:25lat*0:25 long grid as the RCM data." This information will be added to this paper.

- ENSEMBELS data was used because at the time the first part of this research (published in Olsson et al. 2015) was carried out the EURO-CORDEX data was not yet available and we wanted here to use the same data as in the earlier paper. Prein et al. 2015 have concluded that the best largest improvements can be found in regions with substantial orographic features. Finland is relatively flat and therefore the added value of fine resolution is smaller than in some other parts of Europe. Casanueva et al. (2016) found only limited added value of higher resolution in the precipitation frequency and intensity. Text about this will be added to the manuscript.

- Comparison with delta change method will be extended to explore the tradeoffs in

these methods. Also further comparison between articles using several methods will be added.

- Adequacy issue will be further explored for increasing the importance (and novelty) of the paper. New more precise methodology for the assessment of adequacy will be presented and tested. We propose that the performance of the hydrological model and the bias corrected RCM scenarios could be tested separately for snowmelt and rain-induced floods. The estimated 2/10, and 100 year floods and their confidence limits could be compared with the control simulation/observations to testing the performance of the DBS-methods for both frequent and extreme floods.

Specific comments:

These minor corrections will be done in the paper.

Best regards

Noora Veijalainen

Juho Jakkila
Interactive
comment

**Fig. 1.** Comparison of simulated maximum discharges and Gumbel distribution of Hypöistenkoski for snowmelt floods (a and c) and rain induced floods (b and d) in 1961-2000 for bias corrected RCM scenarios.

---

## Author Comment (AC2) · 15 Feb 2018

Dear Reviewer

We would like to thank you for your useful comments on our manuscript. Here are our replies to the comments.

General comments:

The reviewer expressed concern on the choice of the study basins. To address this issue we added the study site of Hypöistenkoski to the study basins to better enable the evaluation of impacts of the method used on floods caused by daily precipitation

on small catchment. Hypöistenkoski is a small river in south-western Finland with 325 km2 catchment area and 1 % lake percentage. The results of Hypöistenkoski (see attached Figure) show that there is no significant difference in the performance of the single gamma and double gamma distribution in correcting of the snowmelt floods as expected. However, the double gamma method corrects the maximum annual rain induced floods better than the single gamma method. Typically the annual maximum floods in Finland have been snowmelt floods, but in the future in southern and coastal part of Finland the annual maximum floods will be more frequently rain induced floods due to climate change. Therefore we will add the results of Hypöistenkoski catchment and discussion about the importance of the performance of the bias correction methods for producing snowmelt and rain induced floods separately.

We will also reformulate the objectives of the study to put more emphasis for the testing of adequacy of the RCM simulations.

1) This section of the manuscript concerning delta change and bias correction methods could indeed be improved and it will be revised including a more comprehensive and up to date literature review.

2) We have looked into the method proposed by Hempel et al. and believe it would indeed help retain the trends in average precipitation. However, as demonstrated by Olsson et al. 2015 (the first part of this study), the trends in average precipitation did not change significantly by either single or double gamma distribution. The trends in extreme precipitation, which changed differently with double gamma distribution than in the uncorrected data, may change differently also in the method proposed by Hempel et al. While this would be an interesting topic to compare, we unfortunately do not have the resources or the space to carry out this comparison in this study and this paper.

3) ENSEMBLES data was used because at the time the first part of this research (published in Olsson et al. 2015) was carried out the EURO-CORDEX data was not yet available and we wanted here to use the same data as in the earlier paper. Prein et

al. 2015 have concluded that the largest improvements with higher resolution can be found in regions with substantial orographic features. Finland is relatively flat country and therefore the added value of fine resolution is smaller than in some other parts of Europe. Casanueva et al. (2016) found only limited added value of higher resolution in the precipitation frequency and intensity. The text regarding the choice of RCM data will be modified and comparison of ENSEMBLES and EURO-CORDEX data will be added.

4) The focus of this article has been in the changes of the extreme floods. The changes in mean high discharges and mean low discharges were studied in previous article (Olsson et al. 2015), but the adequacy of the bias correction method was not tested. In this paper we have proposed to test the adequacy of the bias correction method basing on 95 % confidence limits of the 100 year floods of the control simulation. This test could be improved by extending it to the lower return periods (2-10 years) for testing the performance of the bias correction method in more frequent floods and distribution of the annual maximum floods. The testing of adequacy of the RCMs will be further demonstrated and

5) The likelihood test was applied to the observed discharges and to the control floods (discharges simulated with hydrological model using observed temperature and precipitation). More comprehensive analysis of the GEV vs. Gumbel issue will be added to the manuscript and the changes in the skew parameter of GEV distribution fitted for the simulated floods will be analyzed. In future work the changes for snowmelt floods and rainfall floods should maybe be estimated separately.

Minor comments

The necessary changes will be made to the manuscript. These include:

1) Language of the paper will be checked. 2) Title will be re-evaluated and modified to e.g. Bias correction of Regional Climate Model data in estimation of extreme floods in Finland- comparison of two gamma distributions and evaluation of adequacy of results.

3) Information on the 67 catchments will be added, possibly in a table in an appendix (since these take up quite a lot of space). 4) The sentence will be reformulated. 5) In Olsson et al. 2015: "The areal values of the meteorological observations are calculated for each sub-basin of the hydrological model from three closest observation stations by inverse distance weighting taking into account the elevation differences. The areal values were converted to the same regular 0:25lat*0:25 long grid as the RCM data." This information will be added to this paper. 7) The discussion section will be restructured.

Best regards

Noora Veijalainen

Juho Jakkila

**Snowmelt floods in Hypöistenkoski RCA and HadRM**

**Rain floods in Hypöistenkoski RCA and HadRM**

**Snowmelt floods in Hypöistenkoski HIRHAM A and B**

**Rain floods in Hypöistenkoski**

**Fig. 1.** Comparison of simulated maximum discharges and Gumbel distribution of Hypöistenkoski for snowmelt floods (a and c) and rain induced floods (b and d) in 1961-2000 for bias corrected RCM scenarios.

---

## Editor Comment (EC1) · J. Seibert (Editor) · 21 Feb 2018

This manuscript deals with extreme floods under climate change in Finland. While this is an interesting and timely issue, the two excellent reviews raised important concerns. Main issues include a not fully up-to-date awareness of recent studies, which besides the incomplete review also (and this is the more important point) seemed to have resulted in the use of somewhat outdated methods. Another issue is that the focus and novel aspects are not fully clear. Here it is important to concentrate on and highlight those parts, which are novel contributions. The selection of the catchments is another point that has been criticized.

The responses of the authors indicate that they are willing and able to revise the manuscript, and if the reviewer comments are taken seriously, I am confident that this can become a valuable contribution for HESS.

Best regards, Jan Seibert